# Deep Learning on Histopathological Images for Colorectal Cancer Diagnosis: A Systematic Review

**DOI:** 10.3390/diagnostics12040837

**Published:** 2022-03-29

**Authors:** Athena Davri, Effrosyni Birbas, Theofilos Kanavos, Georgios Ntritsos, Nikolaos Giannakeas, Alexandros T. Tzallas, Anna Batistatou

**Affiliations:** 1Department of Pathology, Faculty of Medicine, School of Health Sciences, University of Ioannina, 45500 Ioannina, Greece; abatista@uoi.gr; 2Faculty of Medicine, School of Health Sciences, University of Ioannina, 45500 Ioannina, Greece; faybirbas@gmail.com (E.B.); kanavosus@gmail.com (T.K.); 3Department of Hygiene and Epidemiology, Faculty of Medicine, University of Ioannina, 45500 Ioannina, Greece; gntritsos@uoi.gr; 4Department of Informatics and Telecommunications, University of Ioannina, 47100 Arta, Greece; tzallas@uoi.gr

**Keywords:** colorectal cancer, CRC, histopathology, microscopy images, deep learning, DL, convolutional neural networks, CNN

## Abstract

Colorectal cancer (CRC) is the second most common cancer in women and the third most common in men, with an increasing incidence. Pathology diagnosis complemented with prognostic and predictive biomarker information is the first step for personalized treatment. The increased diagnostic load in the pathology laboratory, combined with the reported intra- and inter-variability in the assessment of biomarkers, has prompted the quest for reliable machine-based methods to be incorporated into the routine practice. Recently, Artificial Intelligence (AI) has made significant progress in the medical field, showing potential for clinical applications. Herein, we aim to systematically review the current research on AI in CRC image analysis. In histopathology, algorithms based on Deep Learning (DL) have the potential to assist in diagnosis, predict clinically relevant molecular phenotypes and microsatellite instability, identify histological features related to prognosis and correlated to metastasis, and assess the specific components of the tumor microenvironment.

## 1. Introduction

Colorectal cancer (CRC) is one of the most common types of gastrointestinal cancer, the second most common cancer in women and the third in men [1]. Despite existing variations, such as geographical distribution, age and gender differences, the CRC incidence, overall, is estimated to increase by 80% in the year 2035, worldwide [2]. This rising incidence of CRC is mainly due to changes in lifestyle, particularly dietary patterns [3]. Most CRCs are sporadic (70–80%), while approximately one third have a hereditary component [4]. Within the term CRC, a wide range of carcinoma subtypes is included, characterized by different morphological features and molecular alterations.

The cornerstone of CRC diagnosis is the pathologic examination (biopsy or surgical excision) [5]. With the advent of screening methods, many precursor lesions are also detected and biopsied. Consequently, a wide range of pre-malignant lesions have been identified, and occasionally, a differential diagnosis between pre-malignant and malignant lesions is quite challenging [6]. The histopathological examination of the tissue remains the “gold standard” for diagnosis, with the first step being the optimal preparation of the histological section, stained with Hematoxylin and Eosin (H&E) [7]. Further examination with special in situ methods, such as immunohistochemistry (IHC) and in situ hybridization (ISH), and other molecular techniques follows [8]. There are published guidelines for pre-analytical, analytical and post-analytical procedures in a pathology laboratory [9]. As expected, due to the high incidence of CRC, the diagnostic load in a routine pathology laboratory is very heavy and the introduction of an ever-growing list of morpho-molecular features to be examined and noted has made the diagnosis a time-consuming process [10]. All these factors, in combination with the shortage of pathologists worldwide, have led to delays in diagnosis, with consequences to the optimal healthcare of the patient.

It has been shown that pathologists make a diagnosis based mainly on image-based pattern recognition [6]. With this strategy, architectural and cellular characteristics conform to already known features of a disease [11]. In several instances, an accurate diagnosis or estimation of prognostic and predictive factors is subject to personal interpretations, leading to inter- and intra-observer variability [12,13]. In a continuous effort to improve the accuracy of the pathology diagnosis, combined with the timely delivery of all vital information for optimal patient treatment, the new and breakthrough technologies can be of great value. Thus, in the last 5 years, the development of reliable computational approaches, using machine learning based on pattern recognition, has exponentially increased, as reflected in the plethora of published papers [14,15].

The recent World Health Organization (WHO) classification for malignant epithelial tumors of the colorectum includes four main categories: adenocarcinoma (ADC) not otherwise specified (NOS), neuroendocrine tumor NOS, neuroendocrine carcinoma NOS and mixed neuroendocrine-non-neuroendocrine neoplasm (MiMEN) [16]. Of these, colorectal ADC is the most common (90%) and, by definition, it shows glandular and mucinous differentiation. Colorectal ADC has several histopathological subtypes, with specific morphologic, clinical, and molecular characteristics, i.e., serrated ADC, adenoma-like ADC, micropapillary ADC, mucinous ADC, poorly cohesive carcinoma, signet-ring cell carcinoma, medullary ADC, adenosquamous carcinoma, carcinoma undifferentiated NOS and carcinoma with sarcomatoid component.

The diagnosis of CRC is only the first step for a complete pathology report. According to best-practice guidelines, the specific histologic subtype, the histologic grade, the TNM staging system, the lymphovascular and perineural invasion, and the tumor budding should be reported [9,16]. In recent years, the molecular pathological classification of CRC has been proposed, aiming to compliment the traditional histopathologic classification [4]. An integrated molecular analysis performed by the Cancer Genome Atlas Network, has classified CRC into three groups, including highly mutated tumors (~13%), ultra-mutated tumors (~3%) and chromosomal instability (CIN) tumors (~84%). In 2015, an expression-signature-based classification was proposed with four consensus molecular subtype (CMS) groups: CMS1 subtype (MSI-immune, 14%), CMS2 subtype (canonical, 37%), CMS3 subtype (metabolic, 13%) and CMS4 subtype (mesenchymal, 23%). In addition, molecular alterations are prevalent in CRC, consisting of Chromosomal Instability (CIN), Microsatellite Instability (MSI) and a CpG Island Methylator phenotype (CIMP). Defective mismatch repair (MMR) DNA mechanisms lead to increased mutations and, consequently, to MSI [17,18]. The majority of sporadic CRCs are characterized by CIN (~84%), and ~13–16% are hypermutated with an MSI status. The immunohistochemical detection of either an abnormal expression or a loss of expression of the mismatch repair proteins, MLH1, MSH2, MSH6, and PMS2, is of significant diagnostic and prognostic value in CRC, as well as for the detection of hereditary nonpolyposis colorectal cancer (HNPCC), also known as Lynch syndrome, which constitutes approximately 2% to 3% of all colorectal carcinomas [19,20,21].

Histopathology image generations start with the standard procedure of tissue preparation. Biopsy or surgical specimens (representative sections) are formalin-fixed and paraffin-embedded. Then, the 4μm tissue sections are prepared and stained with H&E dye [22]. The images are extracted after a scanning procedure. Several scanning systems can be used to digitize the whole slide [23], such as the Hamamatsu NanoZoomer series, the Omnyx scanner, the Zeiss scanners, the Pannoramic 250 Flash II, and the Leica Biosystems Aperio systems [24]. Most of the above scanners provide two optical magnifications, 20× and 40×, however, the user can also digitally undersample the image in different magnifications. A scanner needs several minutes for the scanning of the whole slide, while most of the system can deal with tens or hundreds of slides that are scanned automatically one-by-one. According to the digitalization, each pixel of a Whole Slide Image (WSI) corresponds to a physical area of several decades nm^2^. For example, in the 40× magnification mode, a Hamamatsu NanoZoomer scanner extracts an image, where the size of each pixel edge corresponds to 227 nm [25]. The latter image digitization provides an appropriate resolution for most of the histological findings, which presents a physical size of microns [26]. In most of the cases, the extracting images are storage either in a compressed JPEG-based format or an uncompressed TIFF format. Figure 1 presents the resolution of a WSI, scanned by a Hamamatsu NanoΖoomer 210.

Machine learning is a branch of AI which is based on the concept that machines could have access to data and be able to learn on their own. AI has a broader scope and involves machines that are capable of carrying out tasks requiring intelligence. Machine learning techniques focus on the creation of intelligent software using statistical learning methods and require access to data for the learning procedure [27]. A branch of machine learning, which has drawn a lot of attention over the last few years, is DL. DL involves training artificial neural networks (ANNs) with multiple layers of artificial neurons (nodes). Neural networks are inspired from the human physiology of the brain, comprising a simplified artificial model of the human neural network. An ANN is a collection of connected artificial neurons. The simplest ANN architecture is the single layer feed forward neural network. In these types of networks, the information moves in one direction only, from the inputs’ nodes to the hidden layer nodes and then to the output nodes. The success and wide acceptance of ANNs relies on their capability to solve complex mathematical problems, nonlinear or stochastic, by using very simple computational operations. In contrast to a conventional algorithm, which needs complex mathematical and algorithmic operations and could only apply to one problem, an ANN is computationally and algorithmically very simple and its structure allows it to be applied in a wide range of problems [28].

DL has rapidly developed during the last decade due to the significant increase in processing power and to the fact that, for the first time, artificial models are able to achieve more accurate results than humans in classification tasks [29]. Both DL and machine learning techniques in general affect our everyday life in various ways. From the simple-looking face recognition program used in Facebook, to the classification of abnormal/normal human cells in bioinformatics. For image analysis problems, such as the histological lesions’ detections, prognosis and diagnosis, DL approaches mainly employ Convolutional Neural Networks (CNNs) for segmentation and classification, while few studies employ another DL approach, called Generative Adversarial Networks (GANs), to improve the training set of images before classification.

CNNs have produced high classification rates in modern computer vision applications. The term “convolutional” suggests that a deep neural network applies the mathematical convolution operation to at least one of its multiple hidden layers. Many CNN model variations have been implemented in recent years, which are based on a common layer pattern: (a) 1 input layer, (b) L-1 convolution layers and (c) 1 classification layer. The key feature of a sequential CNN is that it transforms the input data through neurons that are connected to neurons of the previous convolution layer. Initially, the raw image is loaded at the input layer, which is usually set to accept a three-dimensional spatial form of an image file (width × height × depth), with the depth, in this case, indicating the RGB (Red, Green, Blue) color channels. More technically, each of the convolution layers calculates the dot product between the area of the neurons in the input layer and the weights in a predetermined size of a filtering kernel (e.g., 3 × 3). In this way, local features can be detected through K declared kernels. As a result, all nodes (neurons) of each convolution layer calculate their activation value based on only one subset of spatially adjacent nodes on the filtered feature maps of each previous convolution layer. The most common deep network architectures, such as AlexNet and GoogleNet, use the same neuron type at each hidden layer [30,31]. These architectures achieve very high accuracy in classification problems, while their training is a computationally intensive and time-consuming process. Currently, many different architectures, such as VGG, DenseNet, ResNet, Inception.v3, etc., have been proposed, performing well under different conditions and problem parameters [31,32,33].

GANs are also a DL approach applied on digital image analysis [34]. GANs are a smart way to train a model as a supervised learning problem, even if based on their principles they are unsupervised machine learning procedures. A typical GAN consists of two sub-models: (a) the generator network, where the training generates new samples with similar characteristics to the real ones and (b) the discriminator network, which provides a binary classification of the generating samples, discriminating the real (approved) samples from the fake ones. GANs have been rapidly evolved, especially in image processing and classification, providing a sophisticated approach to simulate images for CNN training, avoiding overtraining and overfitting. It is an alternative method of image augmentation which extracts simulated images using simple transformations such as rotation, shearing, stretching, etc.

In this paper, a systematic review for the application of DL in colorectal cancer, using digital image analysis in histopathological images, is presented. The aim of the manuscript focuses on the investigation from both medical and technical viewpoints. The innovative contribution of this systematic review is the combination of the two viewpoints provided, presenting a more comprehensive analysis of AI-based models in CRC diagnosis. A deeper understanding on both medical and technical aspects of DL will better reveal the opportunities of implementing DL-based models in clinical practice, as well as overcome several challenges occurring for the optimal performance of the algorithms. According to the PRISMA guidelines [35], an expanded algorithm was used for searching the literature works. Specific inclusion and exclusion criteria have been defined to result in the final studies of interest, which have been categorized for both medical and technical points of views. In the next sections, significant backgrounds for both the clinical practice and the details about DL in image analysis are outlined, the method for the study selection is analyzed, and results are extensively discussed.

## 2. Materials and Methods

### 2.1. Search Strategy 

We systematically searched PubMed from inception to 31 December 2021 for primary studies developing a DL model for the histopathological interpretation of large bowel biopsy tissues and CRC. For this purpose, we used the following algorithm: (convolutional neural networks OR CNN OR deep learning) AND ((cancer AND (colon OR colorectal OR intestin* OR bowel)) OR (adenocarcinoma AND (colon OR colorectal OR intestin* OR bowel)) OR (carcinoma AND (colon OR colorectal OR intestin* OR bowel)) OR (malignan* AND (colon OR colorectal OR intestin* OR bowel))) AND (biop* OR microscop* OR histolog* OR slide* OR eosin OR histopatholog*). The search was conducted on 14 January 2022. 

### 2.2. Study Eligibility Criteria

The study was conducted according to the PRISMA guidelines and registered to PROSPERO 2020. Eligible articles were considered based on the following criteria. We included studies presenting the development of at least one DL model for the histopathological assessment of large bowel slides and CRC. Eligible applications of the DL models included diagnosis, tumor tissue classification, tumor microenvironment analysis, prognosis, survival and metastasis risk evaluation, tumor mutational burden characterization and, finally, microsatellite instability detection. We excluded articles that presented in vitro models, used endoscopic or radiological images instead of histological sections, and involved non-photonic microscopy. Furthermore, eligible articles should report original studies and not reviews/meta-analyses, concern humans and be written in English. Additionally, articles referring to organs other than the large bowel and benign entities were deemed ineligible.

### 2.3. Study Selection

All citations collected by the previously mentioned methodology were independently screened by four researchers, who were properly trained before the process started, using the online software Rayyan. Three of the researchers were scientifically capable of evaluating the medical aspect of the query and one of them was a CNN expert, able to assess the technical part. During the screening period, the researchers would meet regularly to discuss disagreements and continue training. Conflicts were resolved by consensus. The full texts of potentially eligible articles were later retrieved for further evaluation.

### 2.4. Data Extraction

To facilitate the data extraction process, we specially designed a spreadsheet form, which all researchers could access to import data from all the eligible articles. From each paper, we extracted information on first author, year and journal of publication, PubMed ID, title, aim of medical research, technical method, classification details, dataset and performance metrics.

## 3. Results

Our systematic search returned 166 articles, 92 of which were selected for full-text screening. Finally, 82 articles were considered eligible for our systematic review according to our criteria of eligibility. A detailed description of the study selection process can be found in the PRISMA flow-chart presented in Figure 2. The selected works are presented both through the medical and technical point of view (Figure 3), while Table 1 includes the characteristics of each study, regarding the medical scope, the technical approach, the employed datasets, and finally, the performance of the proposed method.

### 3.1. Medical Viewpoint

According to the medical scope of view, there are five categories: (a) studies for diagnostic purposes, (b) the classification of the tumor tissue, (c) the investigation of the tumor microenvironment, (d) the role of histological features to prognosis, metastasis and survival, and finally, (e) the identification of microsatellite instability.

#### 3.1.1. Diagnosis

DL techniques can assist in the process of pathology diagnosis [14]. The algorithms perform a binary classification, for instance, cancer/non-cancer, colon benign tissue/colon ADC.

The classification of the tumor regions in WSIs by AI-based models could assist in the time-consuming process of a microscopical examination. The suggested models in the study by Gupta et al. classified normal and abnormal tissue in CRC slides and localized the cancer regions with good performance metrics [36]. Zhou et al. used global labels for tumor classification and localization without the need for annotated images [37]. In the same framework, DL algorithms performed a binary classification of CRC images for detecting cancerous from non-cancerous regions, achieving good performance metrics and supporting the potential for use in clinical practice [38,39,40,41,42]. A recent study evaluating the segmentation performance of different DL models, showed that AI-patch-based models had great advantages, although this segmentation approach could result in lower accuracy when more challenging tumor images are included [43]. Moreover, AI-based models could be combined to persistent homology profiles (PHPs) and effectively identify normal from tumor tissue regions, evaluating the nuclear characteristics of tumor cells [44]. A patch-cluster-based aggregation model, including a great number of WSIs developed by Wang et al., performed the classification of CRC images (cancer, not cancer) assessing the clustering of tumor cells, and the results were comparable to pathologists’ diagnosis, revealing no statistical difference [45]. The acceleration of tumor detection by CNNs could be obtained by reducing the number of patches, taking care to select the most representative regions of interest [46]. Both proposed methods in the study of Shen et al. performed with good accuracy and efficiency in detecting negative cases. Lastly, Yu et al., using a large dataset, demonstrated that SSL, with large amounts of unlabeled data, performed well at patch-level recognition and had a similar AUC as pathologists [47].

Colon benign tissue and colon ADC were classified with good accuracy by DL models developed by Toğaçar et al. and Masud et al. [48,49]. The study of Song et al. showed that the DL model and the pathologists’ estimation were in agreement in diagnosing CRC [50]. However, the binary classification algorithm for adenoma and non-cancerous (including mucosa or chronic inflammation) tiles showed a proportion of false predictions in challenging tiles consisting of small adenomatous glands.

The accurate identification of benign from malignant tissues achieved a sensitivity of 0.8228 and specificity of 0.9114 by a DL model trained with Multiphoton microscopy (MPM) images, although images were lacking biomarkers such as colonic crypts and goblet cells [51]. Holland et al. used the same classification model and 7 training datasets consisting of a descending number of images [52]. The mean generalization accuracy appeared to rely on the number of images within the different training sets and CNNs, although the larger datasets did not result in a higher mean generalization accuracy, as expected.

#### 3.1.2. Tumor Tissue Classification (Non-Neoplastic, Benign, Malignant, Grade, Architecture and Cellular Characteristics)

Lizuka et al. conducted a classification of CRC into adenocarcinoma, adenoma or normal tissue on three different test sets, revealing great performance metrics and promising results for clinical practice [53]. The progression of CRC could be assessed by CNN, designed to identify benign hyperplasia, intraepithelial neoplasia, and carcinoma using multispectral images, however, the contribution of the pathologist’s assessment and a bigger dataset were required [54]. Another study demonstrated that colorectal histological images could be classified into normal mucosa, an early preneoplastic lesion, adenoma and cancer with good accuracy, although these four classes may occasionally overlap and result in uncertainty in labeling [55]. Moreover, the ARA-CNN model was designed for an accurate, reliable and active tumor classification in the histopathological slides, aiming to minimize the uncertainty of mislabeled samples [56]. The model achieved great performance metrics not only in the binary, but also in the multiclass tumor classification, such as the proposed CNN by Xu et al. and Wang et al. [57,58]. Three studies by Papadini et al., Jiao et al. and Ben Hamida et al. proposed CNN approaches for multi-class colorectal tissue classification in a large dataset number, underlining the great potential of AI-based methods to efficiently perform multiple classifications of tumor regions [59,60,61]. Repurposing a stomach model trained in poorly differentiated cases of gastric ADC using a transfer learning method, DL algorithms could perform the classification of poorly differentiated adenocarcinoma in colorectal biopsy WSIs, benefiting from histological similarities between gastric and colon ADC [62].

The challenging task of gland segmentation was approached by Xu et al. and Graham et al., developing CNNs for gland segmentation and achieving a good performance in statistical metrics as well as generalization capability [63,64]. In addition, Kainz et al. trained two networks to recognize and separate glands which achieved 95% and 98% classification accuracy in two test sets [65]. Further research, both in H&E-stained and IHC images of colorectal tissue, was performed for glandular epithelium segmentation [66].

Grading into normal, low-grade and high-grade CRC was approached by Awan et al. and Shaban et al. with 91% and 95.7% accuracy, respectively, using the same dataset [67,68]. Lastly, the grading of colorectal images was performed by an unsupervised feature extractor via DL, showing great accuracy, although, as expected, the subcategorization of low-grade tissue images had reduced the accuracy [69].

#### 3.1.3. Tumor Microenvironment

An automated assessment of the CRC tumor microenvironment was carried out, including the stroma, necrosis and lymphocytes associated with progression-free intervals (PFI) [70]. Jiao et al. demonstrated that a higher tumor–stroma ratio was a risk factor, whilst high levels of necrosis and lymphocytes features were associated with a low PFI. Pham’s et al. proposed a DL model for binary and 8-class tumor classification in CRC images, as well as, for the prediction and prognosis of the protein marker, DNp73 in IHC rectal cancer images provided perfect results and outperformed other CNNs [71]. Pai et al. conducted a tumor microenvironment analysis in colorectal TMAs [72]. The algorithm efficiently detected differences between MMRD and MMRP slides based on inflammatory stroma, tumor infiltrating lymphocytes (TILs) and mucin, and the quantified proportion of tumor budding (TB), and poorly differentiated clusters (PDCs) associated with lymphatic, venous and perineural invasion. A Desmoplastic Reaction (DR) could be also classified by DL algorithms in CRC histopathological slides containing the deepest tumor invasion area [73]. The classification of a DR based on a myxoid stroma could be a significant prognostic marker for patients’ survival.

Comprehensive analysis of the tumor microenvironment proved to show a great performance by the ImmunoAIzer, a DL model for cell distribution description and tumor gene mutation status detection in CRC images, proposed by Bian et al. [74]. Optimal results were achieved in accuracy and precision for biomarker prediction, including CD3, CD20, TP53 and DAPI. Additionally, the suggested DL framework could effectively quantify TILs, PD-1 expressing TILs in anti-PD-1 immunofluorescence staining images, as well as detect APC and TP53. Lymphocytes could be detected in colorectal IHC images stained positive for CD3 and CD8 biomarkers by 4 different CNNs, with U-Net showing the best performance according to the F1 score [75]. In the same framework, Xu et al. proposed a DL model for the quantification of the immune infiltration (CD3 and CD8 T-cells’ density) within the stroma region using IHC slides [76]. The CNN-IHC model performed with high accuracy and was efficient in predicting survival probability, which was increased when patients had a higher stromal immune score. Predictions of genetic mutation genes, such as APC, KRAS, PIK3CAM SMAD4, TP53 and BRAF, could be followed through the DL algorithms to support the clinical diagnosis and better stratify patients for targeted therapies [77,78]. Schrammen et al. proposed the Slide-Level Assessment Model (SLAM) for simultaneously tumor detection and predictions of genetic alterations [79]. In a 2017 study, recognizing the molecular tumor subtype based on histopathology image data, Popovici et al. proposed a challenging approach utilizing a DCNN, which was effective in predicting relapse-free survival [80]. Xu et al. compared a DCNN to handcraft feature representation in IHC slides of CRC, stained for an Epidermal Growth Factor Receptor (EGFR), and demonstrated that the DCNN showed a better performance versus the handcrafted features in classifying epithelial and stromal regions [81]. In addition, Sarker et al. developed a DL approach for the identification and characterization of an Inducible T-cell COStimulator (ICOS) biomarker, which achieved high accuracy in the ICOS density estimation and showed potential as a prognostic factor [82]. Tumor budding could be quantified in CRC IHC slides stained for pan-cytokeratin, whereas a high tumor budding score was correlated to a positive nodal status [83].

Analysis for cell nuclei types (epithelial, inflammatory, fibroblasts, “other”) by a CNN model trained on 853 annotated images showed a 76% classification accuracy [26]. All four cell types were associated with clinical variables, for instance, fewer inflammatory cells were related to mucinous carcinoma, while metastasis, residual tumors, as well as venous invasion were related to lower numbers of epithelial cells. A similar study, by Sirinukunwattana et al., described a CNN method for the detection and classification of four cell nuclei types (epithelial, inflammatory, fibroblast and miscellaneous) in histopathological images of CRC [84]. Höfener et al. used the same dataset as Sirinukunwattana et al. for nuclei detection from Cthe NNs based on the PMap approach [85]. A novel CNN architecture, Hover-net, was proposed by Graham et al. for the simultaneous segmentation and classification of nuclei, as well as for the prediction of 4 different nuclear types [86]. In 2017, the deep contour-aware network (DCAN) was developed by Chen et al. for accurate gland and nuclei segmentations on histological CRC images [87].

#### 3.1.4. Histological Features Related to Prognosis, Metastasis and Survival

A peri-tumoral stroma (PTS) score evaluated by CNNs was significantly higher in patients with positive lymph nodes compared to the Lymph Node Metastasis (LNM)-negative group [88]. However, due to the small dataset and the selection of classes used, the PTS score for LNM and extramural tumor deposits in early-stage CRC was not detected. Kiehl et al. and Brockmoeller et al. showed that LNM could be predicted by DL models with a good performance [89,90]. Furthermore, the incidence of metastasis in histologic slides with one or more lymph nodes was predicted by CNN, with good accuracy, both for micro- and macro-metastases [91].

Bychkov et al., using TMAs of the most representative tumor area of CRC, proved the efficiency of a DL model to predict the 5-year disease-specific survival (DSS), while Skrede et al. reported data for the prediction of cancer-specific survival [92,93]. Similarly, DSS was predicted by a DL model and clinicopathological features, such as poorly differentiated tumor cell clusters, were associated with high DL risk scores [94]. A Crohn-like lymphoid reaction (CLR) density at the invasive front of the tumor was a good predictor of prognosis in patients with advanced CRC, independent of the TNM stage and tumor–stroma ratio [95]. Determining the ratio of the desmoplastic and inflamed stroma in histopathological slides by DL models could be of great value in predicting the recurrence of disease after rectal excision and a lower desmoplastic to inflamed stroma ratio was associated with a good prognosis [96]. Tumor–stroma ratio (TSR) measures could be an important prognostic factor and, as shown by Zhao et al. and Geesink et al., a stroma-high score was associated with reduced overall survival [97,98]. The “deep stroma score” by Kather et al., a combination of non-tumor components of the tissue, could be an independent prognostic factor for overall survival, especially in patients with advanced CRC [99]. IHC slides stained for pan-cytokeratin from patients with pT3 and pT4 colon ADC were used to train a DCNN to predict the occurrence of distant metastasis based on tumor architecture [100]. Another study showed that IHC-stained images of the amplified breast cancer 1 (AIB1) protein from CRC patients could operate as a predictive 5-year survival marker [101].

#### 3.1.5. Microsatellite Instability

Deploying the dataset of the MSIDETECT consortium, Echle et al. developed a DL detector for the identification of MSI in histopathological slides [102]. High MSI scores were accompanied by the presence of a poorly differentiated tumor tissue, however, false MSI scores were also noted in necrotic and lymphocyte infiltrated areas. The binary classification of DL algorithms for predicting MSI and MSS status in CRC images was performed in studies by Wang, Yamashita, Bustos and Cao et al., with the latter study associating MSI with genomic and transcriptomic profiles [103,104,105,106]. Another MSS/MSI-H classifier model was trained on tumor-rich patch images for better classification results, although some images were misclassified indicating that a larger dataset was required [107]. Generating synthesized histology images could also be utilized by DL models for detecting MSI in CRC, as demonstrated by Krause et al. [108]. A synthetic dataset achieved an almost similar AUC in predicting MSI compared to real images, although the best performance was noted when a combination of synthetic and real images was generated. Image-based consensus molecular subtype (CMS) classification in CRC histological slides from 3 datasets showed a good performance, and the slides having the highest prediction confidence were in concordance with the histological image features [109]. In another study, CMS classification was associated with mucin-to-tumor area quantification, and revealed that CMS2 CRC had no mucin and MUC5AC protein expression was an indication for worse overall survival [110]. Lastly, a CNN for predicting tumor mutational burden-high (TMB-H) in H&E slides was developed by Shimada et al. and showed an AUC of 0.91, while high AUC scores were also noted in the validation cohorts [111]. TMB-H was associated with TILs, although further development is important for this CNN model to be included in clinical practice.

### 3.2. Technical Viewpoint

The presented DL methods for image analysis in colorectal histopathology images could follow a categorization close to the one presented, which is presented in the background section. The systematic review indicates a rapid implementation of the field, presenting DL applications that cover many technical approaches. Most of the presented works in the literature employ a Convolution Neural Network in different segmentation and classification problems (i.e., binary classification for the diagnosis or prognosis of cancer, multiclass problems to characterize different tissue types, segmentation problems for the detection of the microenvironment of the tissue). According to the scope of the study, the authors proposed an appropriate architecture, providing the performance of their method and perhaps comparing with other already developed CNNs. Few studies used GANs to improve the training of the network, while several of them extended architectures for encoding and decoding, such as U-Net. Recent studies took the advantage of a high classification performance, developing retrospective or cohort studies based on the DL results. Technically, almost all the studies utilized popular machine learning environments, such as PyTorch, TensorFlow, Keras, Fastai, etc., which provided robust implementations of DL approaches. The main category of CNN application can be divided into three subcategories: (i) custom CNN architectures, (ii) popular architectures with transfer learning, and finally, (iii) novel architectures, ensemble CNNs or frameworks.

#### 3.2.1. Custom CNN Architecture

Custom CNN architectures denote those approaches where the authors built, from scratch, all the layers of the network, visualizing in detail the feature extraction layers, the fully connected layers of the classifier, as well as all the layers between of them. Commonly, these architectures consisted of few layers and a small number of parameters, instead of the well-known architectures where the networks expanded and were deeper than custom ones. In several cases, custom CNNs performed well for typical simple problems, where it was probably meaningful to avoid complex architectures and networks with a high consuming computational effort. Several proposed custom CNNs were constructed, containing up to 4 convolution layers for feature extraction and up to 2 fully connected layers for the classifier [38,45,53,66,80]. For example, one of the first presented methods by Xu et al. classified the regions of the image as the epithelium or stroma, employing a simple CNN within a total of 4 layers (2 convolution and 2 fully connected) [81]. Other research teams implemented deeper architectures than the latter, including at least 8 layers [40,83,98]. For example, one of the most recent studies used a custom architecture of 15 layers (12 convolutional and 3 fully connected) for diagnosis purposes [40]. Finally, the most complex custom CNN, proposed by Graham et al. and called MilD-Net+, provides simultaneous gland and lumen segmentation [64].

#### 3.2.2. Popular Architectures with Transfer Learning

The most comfortable way to apply CNNs on imaging problems is the utilization of the machine learning environments, where researchers can easily call already developed architecture. Such architectures gradually became very popular due to their standard implementation as well as their ability to transfer learning from the training in other datasets. According to the concept of transfer learning, it is less computationally expensive to employ a pre-trained deep network instead of a network with randomly generated weights, even if the training set includes images with different characteristics and classes. As a result, in most of the cases, the popular models were trained on the ImageNet dataset, which contained many images of different sources [27]. The most common pre-trained model used for CRC is based on the VGG architectures. Four of the studies, presented by Zhao et al. [95,97], Xu et al. [76] and Jiao et al. [70], employed the VGG-19, while two of the studies employed the VGG-16 [41,101]. Furthermore, two other studies compared different parameters of the general VGG architecture [38,80]. The second and third mostly used CNN for CRC is the Inception.v3 [39,45,53,77,111], the Resnet (ResNet-50 used by Chuang et al. [91], ResNet-18 used by Kiehl et al. [90] and Bilal et al. [78], and ResNet-34 used by Bustos et al. [105] and Bilal et al. [78]), or the combination of them called the InceptionResNet.v2 [100]. These architectures introduced the Inception and the residual blocks, which made the model less sensitive to overfitting. Interesting approaches [67,70,88] were developed using either the U-Net model, where the initial image was encoded to a low resolution and then decoded, providing images with similar characteristics or the ShuffleNet [80,91,103]. Finally, other well-known models were also used, such as AlexNet [57], the YOLO detector [75], the CiFar Model [25], the DenseNet [73], the MobileNet [94], LSTM [71], Xception [51], the DarkNet [48] and EfficientNetB1 [62].

In the category with the pre-trained popular models, all the comparative works could be included. These studies employed either the well-known models referenced above [36,61], or other models such as GoogleNet [99], SqueezeNet [52] and ResNeXT [43]. Finally, two studies utilized [72] or proposed [103] cloud platforms where the user can fine tune several hyper-parameters of popular pre-trained architectures.

#### 3.2.3. Novel Architectures

Many research teams focus on the technical innovation evaluating their proposed methodologies in colorectal image datasets. The studies of these categories are mostly (a) modifications of popular architectures, (b) combinations of techniques into a framework, or (c) ensemble approaches.

Several modified architectures were the HoVer-Net [64] based on the Preact-ResNet-50, the KimiaNet [112] based on the DenseNet, the architecture proposed by Yamashita et al. [104] based on the MobiledNet, and finally, the modification of the loss functions on the ResNet proposed by Medela et al. [113]. Finally, Bian et al. [74] proposed an CNN based on the Inception.v3, adding several residual blocks.

Several studies engaged a CNN architecture with other sophisticated methods and concepts of artificial intelligence. One of the first attempts in the field was developed by Sirinukunwattana et al., proposing a combination of a custom CNN architectures with the Spatial Constrain Regression [84]. A similar concept developed two custom CNN architectures with PMaps approaches [85]. Chen et al. presented a novel deep contour-aware network for the detection and classification of the nuclei [87]. A Deep Belief Network for feature extraction, followed by the Support Vector Machines for classification, was deployed by Sari et al. [69]. A recent work employed a Deep embedding-based Logistic Regression (DELR), which also used active learning for sample selection strategy [60]. In two other studies, the DenseNet was combined with Monte Carlo approaches [46], while the Inception.v3 was cooperated with Adversarial Learning [109]. Finally, Kim et al. [114] combined the InceptionResNet.v2 with Principal Component Analysis and Wavelet Transform.

Some other research teams combined two or more CNNs on a single framework. Two different approaches combined the VGG architectures with the concept of the ResNet [66,92], while the ARA-CNN, proposed by Raczkowski et al. [56], combined the ReSNet with the DarkNet. Lee et al. [107] proposed a framework of an initial custom architecture followed by the Inception.v3. Furthermore, three frameworks based on the ResNet were developed by Zhou et al. [37]. Shaban et al. [68] developed a novel context-aware framework consisting of two stacked CNNs. Finally, another combination between different architectures, which was presented in the literature, is the DeepLab.v2 with ResNet-34 [50].

In recent years, voting systems are increasingly used for classification purposes. These ensemble approaches engage two or more algorithms, where the prediction of the highest performance finally prevails. The first ensemble pipeline was presented by Cao et al. in 2020, which votes according to the likelihood extracted from ResNet-18 [106]. Nguyen et al. [42,110] proposed an ensemble approach with two CNNs (VGG and CapsuleNet), while Kheded et al. deployed an approach with three CNNs as combination backbones: (a) the U-Net with the ResNet, (b) the U-Net with the InceptionResNet.v2 and (c) the DeepLab.v3 with Xception [115]. Another ensemble framework was developed by Skrede et al. [93], with ten CNN models based on the DoMore.v1. The most extended voting systems were presented by Paladini et al. [59], who introduced two ensemble approaches using the ResNet-101, ResNeXt-50, Inception-v3 and DenseNet-161. In the first one, called the Mean-Ensemble-CNN approach, the predicted class of each image was assigned using the average of the predicted probabilities of the four trained models, while in the second one, called the NN-Ensemble-CNN approach, the deep features corresponding to the last FC layer are extracted from the four trained models.

#### 3.2.4. Improving Training with GANs

Apart for the segmentation and classification, DL in CRC has also been applied for the improvement of the training dataset using GANs. There have been three works with GANs’ applications presented during the past two years. In the first attempt [108], a Conditional Generative Adversarial Network (CGAN), consisting of six convolution layers for both the generator and the discriminator network, was employed to train the ShuffleNet for the classification. Finally, a very recent study presented a novel GAN architecture, called SAFRON [116], which enabled the generation of images of arbitrarily large sizes after training on relatively small image patches.

**Table 1 diagnostics-12-00837-t001:** Deep learning methods on histopathological images for colorectal cancer diagnosis.

**Year**	First Author	Journal	Aim of Medical Research	Technical Method	Classification Details	Dataset	Performance Metrics
2016	Sirinukunwattana [84]	IEEE Trans Med Imaging	Detection and classification of nuclei	Custom CNN architecture (7-versions) based on the spatially Constrain Regression (a priori)	4-class: epithelial, inflammatory, fibroblast, miscellaneous	>20,000 annotated nuclei from 100 histology images from 10 WSIs	Detection Precision: 0.781, Recall: 0.827, F1 score: 0.802, Classification F1 score: 0.784, AUC: 0.917, Combined detection and classification F1 score: 0.692
	Xu [81]	Neurocomputing	Classification of epithelial and stromal regions	Custom Simple CNN Architecture with 4 Layers (2 × CL and 2 FC) with SVM	Binary (epithelium/stroma)	1376 IHC-stained images of CRC	Classification F1 score: 100%, ACC: 100%, MCC: 100%
2017	Chen [87]	Med Image Anal	Detection and classification of nuclei	Custom CNN: Novel deep contour-aware network	Binary (bening/malignant)	(1) 2015 MICCAI Gland Segmentation Challenge, Training 85 Images Testing 80, (2) 2015 MICCAI Nuclei Segmentation Challenge: Training 15 Images, Testing 18 images	Detection results (MICCAI Glas): F1 score = 0.887, DICE index 0.868 Hausdorff = 74.731 Segmentation results: D1 and D2 metrics from Challenge
	Popovici [80]	Bioinformatics	Prediction of molecular subtypes	VGG-f (MatConvNet library)	5-class: subtypes (Budinská et al., 2013) Μolecular subtypes (denoted A-E)	PETACCURACY:3 clinical trial (Van Cutsem et al., 2009) 300 H/E images	ACC: 0.84, Confusion metrics Precision and Recall per class
	Xu [63]	IEEE Trans Biomed Eng	Classification of Glands	Custom architecture: 3 channel fusions, one based on Faster R-CNN and two based on VGG-16	Binary (bening/malignant)	2015 MICCAI Gland Segmentation Challenge, Training 85 Images Testing 80 Images	Detection results (MICCAI Glas): F1 score (0.893 + 0.843)/2, DICE index (0.908 + 0.833)/2, Hausdorff (44.129 + 116.821)/2
	Haj-Hassan [54]	J Pathol Inform	Tumor tissue classification	Custom Simple CNN (2CL and 1FC), with or without initial segmentation	3-class: benign hyperplasia, intraepithelial neoplasia, carcinoma	CHU Nancy Brabois Hospital: 16 multispectral images	Dice and Jaccard with std for segmentation ACC: 99.17%
	Xu [57]	BMC Bioinformatics	Tumor tissue classification	Alexnet—SVM (shared by the Cognitive Vision team at ImageNet LSVRC 2013)	(1) Binary (cancer/not cancer (2) 6-class: normal (N), ADC, mucinous carcinoma (MC),serrated carcinoma (SC), papillary carcinoma (PC), cribriform comedo-type adenocarcinoma (CCTA)	2014 MICCAI 2014 Brain Tumor Digital Pathology Challenge and CRC image dataset (1) Total 717 H/E Total 693	ACC: (1) Binary: 98% (2) Multiclass: 87.2%
	Jia [41]	IEEE Trans Med Imaging	Diagnosis	3 Stage VGG-16 (publicly available Caffe toolbox)	(1) Binary (Cancer/non cancer) (2) Binary: TMAs (Cancer/non-Cancer)	(1) 330/580 images (CA/NC) (2) 30/30 images (CA/NC)	(2) ODS: 0.447, F-measure: 0.622 (CA), 0.998 (NC)
	Kainz [65]	PeerJ	Classification of Glands	2 × custom CNNs (4 × CL, 2 × FC)	4-class (benign, benign background, malignant, malignant background) add background for each class of the challenge	2015 MICCAI Gland Segmentation Challenge, Training 85 Images (37 benign and 48 malignant). Testing 80 (37/43)	Detection results (MICCAI Glas): F1 score = (0.68 + 0.61)/2, DICE index (0.75 + 0.65)/2, Hausdorff (103.49 + 187.76)/2
	Awan [67]	Sci Rep	Grading of CRC	UNET-based architecture	(A) Binary (normal/cancer) (B) 3-class: normal/low grade/high grade	38 WSIs, extracted 139 parts (71 normal, 33 low grade, 35 high grade)	(A) Binary ACC: 97% (B) 3-vlass ACC: 91%
	Wang [58]	Annu Int Conf IEEE Eng Med Biol Soc	Tumor tissue classification	Simple architecture consisting of 1 CL and 1 FC, which is simultaneously operated in both decomposed images	8-class: tumor epithelium, simple stroma, complex stroma, immune cells, debris, normal mucosal glands, adipose tissue, background	University Medical Center Mannheim 1.000 images	ACC: 92.6 ± 1.2
2018	Bychkov [92]	Sci Rep	Survival	VGG-16 followed by a recurrent ResNet	Binary (low/high risk 5-year disease-specific survival)	Helsinki University Central Hospital, 420 ΤΜAs	Hazard Ratio: 2.3; CI 95%: 1.79–3.03, AUC 0.69
	Eycke [66]	Med Image Anal	Tumor tissue classification/IHC biomarkers quantification	VGG-based architecture including residual units	Binary (bening/malignant)	2015 MICCAI Gland Segmentation Challenge, Training 85 Images Testing 80 (37/43)	Detection results (MICCAI Glas): F1 score = (0.895 + 0.788)/2, DICE index (0.902 + 0.841)/2, Hausdorff (42.943 + 105.926)/2
	Weis [83]	Diagn Pathol	Evaluation of tumor budding	Custom architecture consisting of 8 layers	Binary (Tumor bud/no tumor)	HeiData Training dataset 6292 images, 20 IHC pan-cytokeratin WSIs	R2 value: 0.86
	Höfener [85]	Comput Med Imaging Graph	Nuclei detection	2 × Custom CNN architectures based on PMaps approach	No classification, just nuclei detection	Same with Sirinukunwattana et al., >20,000 annotated nuclei from 100 histology images, from 10 WSI	F1 score of 22 different configurations of CNNs Best F1 score: 0.828
	Graham [64]	Med Image Anal	Diagnosis	Custom complex architecture, named Mild-net	Binary (bening/malignant)	(1) MICCAI Gland Segmentation Challenge, (2) same as Awan et al., 2017	(1) F1 socre: (0.914 + 0.844)/2, Dice: (0.913 + 0.836)/2, Hausdorff (41.54 + 105.89)/2 (2) F1 score: 0.825, Dice: 0.875, Hausdorff: 160.14
2019	Yoon [38]	J Digit Imaging	Diagnosis	6 VGG-based approaches	Binary (normal/Cancer)	Center for CRC, National Cancer Center, Korea, 57 WSIs, 10.280 patches	ACC: 93.48%, SP: 92.76%, SE: 95.1%
	Sari [69]	IEEE Trans Med Imaging	Grading of CRC	Feature Extraction from Deep Belief Network and classification employing linear SVM, Comparison with Alexnet, GoogleNet, Inceptionv3, and autoencoders	(1) 3-class: normal (N), Low Grade (LG), High Grade (HG) (2) 5-class: Normal, Low (1), Low (1–2), Low (2), High	(1) 3236 images 1001 N, 1703 LG, 532 HG) (2) 1468 images	(1) mean ACC: 96.13 (2) mean ACC: 79.28
	Kather [99]	PLoS Med	Prediction of survival	5 different well-known architectures pre-trained with ImageNet (1) VGG-19, (2) AlexNet, (3) SqueezeNet, (4) GoogleNet, (5) ResNet	9-class: adipose tissue, background, debris, lymphocytes, mucus, smooth muscle, normal colon mucosa, cancer-associated stroma, CRC epithelium/survival predictions	(1) NCT, UMM 86 WSIs (100.000 patches) (2) 25 WSIs DACHS (3) 862 WSIs TCGA WSIs (4) 409 WSIs DACHS	9-class: ACC: 94–99%
	Geessink [98]	Cell Onocol	Quantification of tumor–stroma ratio (TSR) for prognosis	Custom architecture proposed by Ciombi et al., 2017 (not included by our search)	9-class: tumor, intratumoral stroma, necrosis, muscle, healthy epithelium, fatty tissue, lymphocytes, mucus, erythrocytes	Laboratory for Pathology Eastern Netherlands 74 WSIs	Overall ACC: 94.6%
	Shapcott [26]	Front Bioeng Biotechnol	Classification of nuclei	CNN based on Tensorflow “ciFar” model	4-class: epithelial/inflammatory/ fibroblast/other	853 images, 142 TCGA images	Detection ACC: 65% Classification ACC: 76%
	Qaiser [44]	Med Image Anal	Diagnosis	Custom architecture with (4 × CL + (ELU), 2FC + Dropout	Binary: tumor/non-tumor	(1) Warwick-UHCW 75 H/E WSIs (112.500 patches), (2) Warwick-Osaka 50 H/E WSIs (75.000 patches)	(A) PHP/CNN: F1 score 0.9243, Precision 0.9267 (B) PHP/CNN: F1 score 0.8273, Precision 0.8311
	Swiderska-Chadaj [75]	Med Image Anal	Detection of lymphocytes	4-different architectures: (1) Custom with 12CL, (2) U-net, (3) YOLLO (based on YOLO detector), (4) LSM (Sirinukunwattana et al. 2016)	3-class: regular lymphocyte distribution/clustered cells/artifacts	28 IHC WSIs	U-Net F1: 0.80 Recall: 0.74 Precision: 0.86
	Graham [86]	Med Image Anal	Classification of nuclei	Novel CNN architecture (named HoVer-Net) based on Preact-ResNet50	4-class: normal, malignant, dysplastic epithelial/inflammatory/miscellaneous/spindle-shaped nuclei (fibroblast, muscle, endothelial)	(1) CoNSeP dataset, 16 WSIs, 41 H/E tiles, (2) Kumar (TCGA) 30 images, (3) CPM-15 (TCGA) 15 images, (4) CPM-17 (TCGA) 32 images, (5) TNBC (Curie Institute) 50 images, (6) CRCHisto 100 images	(1) Dice: 0.853, AJI: 0.571, DQ: 0.702, SQ: 0.778, PQ: 0.547, (2) Dice: 0.826, AJI: 0.618, DQ: 0.770, SQ: 0.773, PQ: 0.597, (4) Dice: 0.869, AJI: 0.705, DQ: 0.854, SQ: 0.814, PQ: 0.697
	Rączkowski [56]	Sci Rep	Tumor tissue classification	Novel architecture (named ARA-CNN), based on ResNet and DarkNet	(A) Binary: tumor/stroma (B) 8-class: tumor epithelium, simple stroma, complex stroma, immune cells, debris, normal mucosal glands, adipose tissue, background	5000 patches (same as Kather et al., 2016)	(1) AUC 0.998 ACC: 99.11 ± 0.97% (2) AUC 0.995 ACC: 92.44 ± 0.81%
	Sena [55]	Oncol Lett	Tumor tissue classification	Custom CNN (4CL, 3FC)	4-class: normal mucosa, preneoplastic lesion, adenoma, cancer	Modena University Hospital, 393 WSIs	ACC: 81.7
2020	Iizuka [53]	Sci Rep	Tumor tissue classification	(1) Inception v3, (2) also train an RNN using the features extracted by the Inception	3-class: adenocarcinoma/adenoma/non-neoplastic	Hiroshima University Hospital, Haradoi Hospital, TCGA, 4.036 WSIs	(1) AUC: (ADC: 0.967, Adenoma: 0.99), (2) AUC: (ADC: 0.963, Adenoma: 0.992)
	Shaban [68]	IEEE Trans Med Imaging	Grading of CRC	Novel context-aware framework, consisting of two stacked CNNs	3-Class: Normal, Low Grade, High Grade	Same as Awan et al., 2017 30000 patches	ACC: 95.70
	Holland [52]	J Pathol Inform	Diagnosis	(1) ResNet (Turi Create library framework), (2) SqueezeNet (Turi Create library framework), (3) AlexNet (TensorFlow)	Binary (benign/malignant)	10 slides, 1000 overlapping images	(1) ResNET: ACC: 98%, (2) AlexNet: ACC: 92.1% (3) SqueezeNet: ACC: 80.4%
	Echle [102]	Gastroenterology	MSI prediction	A modified version of Sufflenet (no details)	Binary (MSI/MSS)	TCGA, Darmkrebs: Chancen der Verhütung durch Screening (DACHS), “Quick and Simple and Reliable” trial (QUASAR), Netherlands Cohort Study (NLCS) QUASAR	Cross-validation cohort: mean AUC 0.92, AUPRC of 0.63 Validation cohort: AUROC 0.95 (without image-preprocessing) and AUROC 0.96 (after color normalization)
	Song [50]	BMJ	Diagnosis	A novel architecture based on DeepLab v2 and ResNet-34. Comparison with ResNet-50, DenseNet, Inception.v3, U-Net and DeepLab.v3	Binary (colorectal adenoma/non-neoplasm)	Chinese People’s Liberation Army General Hospital, 411 slides CJFH and Cancer Hospital, Chinese Academy of Medical Sciences 168 slides	ACC: 90.4, AUC 0.92
	Zhao [98]	EBioMedicine	Quantification of Tumor–stroma ratio (TSR) for prognosis	VGG-19 pre-trained on the ImageNet using transfer learning with SGDM	9-class: Adipose, Background, Debris, Lymphocyte aggregates, Mucus, Muscle, Normal mucosa, Stroma, Tumor epithelium	TCGA-COAD (461 patients), TCGA-READ (172 patients) Same as Kather et al., 2019	Pearson r (for TSR evaluation between CNN and pathologists): 0.939 ICC Mean difference in TSR evaluation between CNN and pathologists: 0.01 Stroma-high vs. stroma-low patients HR (OS): 1.72 (discovery cohort) and 2.08 (validation study)
	Cao [103]	Theranostics	MSI prediction	An ensemble pipeline for the likelihood of each patch, which is extracted from ResNet-18	Binary (MSI/MSS)	TCGA (429 frozen slides), Tongshu Biotechnology Co. (785 FFPE slides)	(a) TCGA-COAD test set: AUC 0.8848 (b) External Validation set: AUC 0.8504
	Xu [39]	J Pathol Inform	Diagnosis	Inception v3 pre-trained on ImageNet	Binary (normal/cancer)	St. Paul’s Hospital, 307 H/E images	Median ACC: 99.9% (normal slides), median ACC: 94.8% (cancer slides) Independent dataset: median ACC: 88.1%, AUROC 0.99
	Jang [77]	World J Gastroenterol	Prediction of IHC biomarkers	(A) Simple CNN architecture for the initial binary problem (B) Inception.v3 for the main classification problem	A) Binary (tissue/no-tissue), B) Binary (normal/tumor), C) Binary (APC, KRAS, PIK3CA, SMAD4, TP53) wild-type/mutation	TCGA 629 WSIs (frozen tissue sections 7 FFPE) Seoul St. Mary Hospital (SMH) 142 WSIs	Frozen WSIs: AUC 0.693–0.809 FFPE WSIs: 0.645–0.783
	Medela [113]	J Pathol Inform	Tumor tissue classification	The authors proposed several different functions. For the evaluation, a ResNet backbone was employed, with modified last layer	8-class: tumor epithelium, simple stroma, complex stroma, immune cells, debris and mucus, mucosal glands, adipose tissue, background	University Medical Center Mannheim, 5.000 H/E images	With K = 3: BAC: 85.0 ± 0.6 Silhouette: 0.37 ± 0.02 Davis–Bouldin: 1.41 ± 0.08 With K = 5: BAC: 84.4 ± 0.8 Silhouette: 0.37 ± 0.02 Davis–Bouldin: 1.43 ± 0.09 With K = 7: BAC: 84.5 ± 0.3 Silhouette: 0.37 ± 0.02 Davis–Bouldin: 1.43 ± 0.09
	Skrede [93]	Lancet	Survival	An ensemble approach with ten different CNN models based on DoMorev1	3-class (good/poor prognosis/uncertain)	>12.000.000 image tiles	Uncertain vs. good prognosis HR: 1.89 unadjusted and 1.56 adjusted Poor vs. good prognosis HR: 3.84 unadjusted and 3.04 adjusted Comparison of 3-year cancer-SP: survival of the good prognosis group to the uncertain and poor prognosis groups: SE: 52%, SP: 78%, PDV:19%, NPV: 94%, ACC: 76% Comparison of 3-year cancer- SP: survival of the good and uncertain prognosis groups with the poor prognosis group: SE: 69%, SP: 66%, PDV: 17%, NPV: 96%, ACC: 67%, AUC: 0.713
2021	Sirinukunwattana [109]	Gut	Consensus molecular subtypes (CMSs) prediction	Inception v3, as well as adversarial learning	4-class: CMS1, CMS2, CMS3, CMS4	(1) FOCUS 510 H/E slides, (2) TCGA 431 H/E slides, (3) GRAMPIAN 265 H/E slides Total: 1.206 slides	(1) AUC 0.88, (2) AUC 0.81, (3) AUC 0.82
	Yamashita [104]	Lancet Oncol	MSI prediction	2-stage novel architecture based on a modified MobileNetV2 architecture pre-trained on ImageNet and fine-tuned by transfer learning on the Stanford-CRC dataset	(1) 7-classes: adipose tissue, necrotic debris, lymphocytes, mucin, stroma or smooth muscle, normal colorectal epithelium, and colorectal ADC epithelium (2) Binary (MSI/MSS)	Stanford-CRC dataset (internal): 66,578 tiles from 100 WSIs TCGA (external): 287,543 tiles from 484 WSIs	Internal: AUROC 0.931, External: AUROC 0.779 NPV:93.7%, SE:76.0%, SP:66.6% Reader study Model AUROC 0.865 Pathologist AUROC 0.605
	Zhou [37]	Comput Med Imaging Graph	Tumor tissue classification	A novel 3-framework based on ResNet. Each framework employs different CNN for (a) Image-level binary classification (CA/NC), (b) Cell-level providing the cancer probability in heatmap, (c) Combination framework which merges the output of the previous ones	Binary (cancer/normal)	TCGA 1346 H/E WSIs, First Affiliated Hospital of Zhejiang University, First Affiliated Hospital of Soochow University, Nanjing First Hospital 50 slides	ACC: 0.946 Precision: 0.9636 Recall: 0.9815 F1 score: 0.9725
	Masud [49]	Sensors	Diagnosis	Custom simple CNN architecture with 3 CL, two max pooling 1 batch normalization and 1 dropout	Binary (Colon ADC/colon benign)	LC25000 dataset, James A. Haley Veterans’ Hospital, 5.000 images of Colon ADC, 5.000 images of Colon Benign Tissue	Peak classification ACC: 96.33% F-measure score 96.38% for colon and lung cancer identification
	Kwak [88]	Front Oncol	Lymph Node Metastasis (LNM) prediction	U-Net based architecture without (no details)	7-class: normal colon mucosa, stroma, lymphocytes, mucus, adipose tissue, smooth muscle, colon cancer epithelium	TCGA 1000.000 patches	LNM positive group/LNM negative group: OR = 26.654 (PTS score) Ability of PTS score to identify LNM in colon cancer: AUC 0.677
	Krause [108]	J Pathol	MSI prediction	A conditional generative adversarial network (CGAN) for synthetic image generation with 6-CL for both the generator and discriminator network, and a modified ShuffleNet for classification	Binary (MSS/MSI)	TCGA (same as Kather et al., 2019) NLCS cohort (same as Echle et al., 2020)	AUROC 0.742 (patient cohort 1), 0.757 (patient cohort 2), 0.743 (synthetic images), 0.777 (both patient cohorts and synthetic images)
	Pai [72]	Histopathology	Tumor microenvironment	CNN developed on the deep learning platform (Aiforia Technologies, Helsinki, Finland) (No details of architecture)	(A) 7-class: carcinoma, tumor budding/poorly differentiated clusters, stroma, necrosis, mucin, smooth muscle, fat (B) 3-class: immature stroma, mature stroma, inflammatory stroma (C) 3-class: low grade carcinoma, high grade carcinoma, signet ring cell carcinoma (D) TILs identification	Stanford University Medical Center (same as Ma et al., 2019) 230 H/E TMAs	MMRD classifying SE: 88% and SP: 73% ICC between pathologists and model for TB/PDCs, type of stroma, carcinoma grade and TILs: 0.56 to 0.88
	Wang [45]	BMC Med	Diagnosis	AI approach uses Inception.v3 CNN architecture with weights initialized from transfer learning	Binary (cancer/not cancer)	14,234 CRC WSIs and 170.099 patches	ACC: 98.11%, AUC 99.83%, SP: 99.22%, SE: 96.99%
	Riasatian [112]	Med Image Anal	Tumor tissue classification	Proposed a novel architecture (called KimiaNet) based on the DenseNet	8-class: tumor epithelium, simple stroma, complex stroma, immune cells, debris, normal mucosal glands, adipose tissue, background	TCGA 5.000 patches	ACC: 96.38% (KN-I) and 96.80% (KN-IV)
	Jiao [70]	Comput Methods Programs Biomed	Tumor microenvironment	(1) For the foreground, tissue detection employs based on U-NET (2) For 9-class problem, employs the same VGG-19 architecture as Kather et al. and Jhao et al.	9-class: adipose tissue, background, debris, lymphocytes, mucus, smooth muscle, normal colon mucosa, cancer-associated stroma, colorectal ADC epithelium	TCGA 441 H/E images	PFI Stroma HR: 1.665 Necrosis HR: 1.552 Lymphocyte HR: 1.512
	Nearchou [73]	Cancers	Classification of Desmoplastic reaction (DR)	DenseNet neural network, integrated within HALO^®^	Binary (Immature/other DR type)	528 stage II and III CRC patients treated at the National Defense Medical College Hospital, Japan	Classifier’s performance: Dice score: 0.87 for the segmentation of myxoid stroma (test set: 40 patient samples)
	Lee [107]	Int J Cancer	MSI prediction	A framework of an initial CNN architecture based on binary classification of patches, followed by an Inception.v3	(A) Binary (tissue/non-tissue) (B) Binary (normal/tumor) (C) Binary (MSS/MSI-H)	TCGA (COAD, READ) 1.336 frozen slides, 584 FFPE WSIs Seoul St. Mary’s Hospital 125 MSS FFPE WSIs, 149 MSI-H FFPE WSIs and 77 MSS FFPE WSIs	TCGA dataset: AUC 0.892 SMH dataset: AUC 0.972
	Wulczyn [94]	NPJ Digit Med	Survival	(1) Tumor segmentation model based on Inception v3, (2) Prognostic model based on Mobile net	Binary (tumor/not tumor)	27.300 slides Validation dataset 1: 9.340 Validation dataset 2:7.140	Validation dataset 1: AUC 0.70 (95% CI: 0.66–0.73) Validation dataset 2: 0.69 (95% CI: 0.64–0.72)
	Shimada [111]	J Gastroenterol	Tumor mutational burden (TMB) prediction	Inception.v3	(A) Binary (neoplastic/non-neoplastic) (B) Binary (TMB-High/TMB-Low)	Japanese cohort TCGA 201 H/E images	AUC 0.910
	Bian [74]	Cancers	Prediction of IHC biomarkers	(1) Modification of Inceptionv3 adding residual block for cellular biomarker distribution prediction and (2) employs Shufflenet.v2, for tumor gene mutation detection	Binary (biomarkers prediction) CD3/CD20, panCK, DAP Binary (tumor mutation genes) APC, TP53, KRAS	Peking University Cancer Hospital and Institute (8697 H/E image patches), TCGA-Colon ADC (COAD) project (50,801 H/E image patches)	Biomarker’s prediction: ACC: 90.4% Tumor gene mutation detection: AUC = 0.76 (APC), AUC = 0.77 (KRAS), AUC = 0.79 (TP53)
	Schiele [100]	Cancers	Survival	InceptionResNet.v2 network, pre-trained on images from the ImageNet from Keras	Binary (low/high metastasis risk)	University Hospital Augsburg 291 pT3 and pT4 CRC patients	AUC 0.842, SP: 79.5%, SE: 75.6%, ACC: 75.8%
	Theodosi [101]	Microsc Res Tech	Survival	Pre-trained VGG-16	Binary (5-year survivors/non-survivors)	University Hospital of Patras 162 IHC AIB1 images	ML system: Mean Overall Classification ACC: 87% DL system: Classification ACC: 97%
	Wang [105]	Bioinformatics	MSI prediction	A platform for automated classification where each user can define his own problem. Different popular architectures have been embedded (Inception-V3, ResNet50, Vgg19, MobileNetV2, ShuffleNetV2, and MNASNET)	Binary (MSI/MSS)	TCGA and WSIs	mean ROC (AUC 0.647 ± 0.029)
	Khened [115]	Sci Rep	Slide Image Segmentation and Analysis	A novel ensemble CNN framework with three pre-trained architectures: (a) U-net with DenceNet as the backbone, (b) U-Net with Inception-ResNet.v2 (Inception.v4), (c) Deeplabv3Plus with Xception	(1) Camelyon16: Binary (normal/metastasis), (2) Camelyon17: 4-class: (negative, ITC, Micro and Macro)	DigestPath 660 H/E images (250 with lesions, 410 with no lesions)	Dice: 0.782
	Chuang [91]	Mod Pathol	Detection of nodal micro- and macro-metastasis	ResNet-50	3-class: Micrometastasis/Macrometastasis/Isolated tumor cells	Department of Pathology, Chang Gung Memorial Hospital in Linkou, Taiwan, 3182 H/E WSIs	Slides with >1 lymph node: Macromatastasis: AUC 0.9993, Micrometastasis: AUC 0.9956 Slides with a single lymph node: Macromatastasis: AUC 0.9944, Micrometastasis: AUC 0.9476 Algorithm ACC: 98.50% (95% CI: 97.75–99.25%)
	Jones [96]	Histopathology	Survival	Νo details for DL	7-class: background, necrosis, epithelium, desmoplastic stroma, inflamed stroma, mucin, non-neoplastic mesenchymal components of bowel wall	Oxford Transanal Endoscopic Microsurgery (TEM) database H/E FFPE 150 patients	For desmoplastic to inflamed stroma ratio: AUC: 0.71, SE: 0.92, SP: 0.50, PPV: 0.30, NPV: 0.97 For stroma to immune ratio: AUC: 0.64, SE: 0.92, SP: 0.45, PPV: 0.27, NPV: 0.96
	Pham [71]	Sci Rep	Tumor tissue classification	Time-frequency, time-space, long short-term memory (LSTM) networks	(1) binary (stroma/tumor), (2) 8-class: tumor, simple stroma, complex stroma, immune cells (lymphoid), debris, normal mucosal glands (mucosa), adipose tissue, background	Colorectal cancer data: University Medical Center Mannheim, 625 non-overlapping for each 8 types of tissue images, total 5000 tissue images	(1) ACC: 100, SE: 100, SP: 100, Precision: 100, F1-score: 1 (2) ACC: 99.96%
	Sarker [82]	Cancers	Prediction of IHC biomarker	U-net architecture with, in total, 23 convolutional layers	Binary (ICOS-positive cell/background)	Northern Ireland Biobank (same as Gray et al., 2017)	U-net highest performance: ACC: 98.93%, Dice: 68.84%, AJI = 53.92% (Backbone: ResNet101, optimizer: Adam, loss function: BCE, batch size: 8)
	Ben Hamida [61]	Comput Biol Med	Tumor tissue classification	(1) Comparison of 4 different architectures Alexnet, VGG-16, ResNet, DenseNet, Inceptionv3, with transfer learning strategy (2) Comparison of SegNet and U-Net for semantic Segmentation	(A) 8-class: tumor, stroma, tissue, necrosis, immune, fat, background, trash (B) Binary (tumor/no-tumor)	(1) AiCOLO (396 H/E slides), (2) NCT Biobank, University Medical Center Mannheim (100.000 H/E patches), (3) CRC-5000 dataset (5.000 images), (4) Warwick (16 H/E)	(1) ResNet On AiCOLO-8: overall ACC: 96.98% On CRC-5000: ACC: 96.77% On NCT-CRC-HE-100κ: ACC: 99.76% On merged: ACC: 99.98% (2) On AiCOLO-2 UNet: ACC: 76.18%, SegNet: ACC:81.22%
	Gupta [36]	Diagnostics	Tumor tissue classification	(a) VGG, ResNet, Inception, and IR-v2 for transfer learning, (b) Five types of customized architectures based on Inception-ResNet-v2	Binary (normal/abnormal)	Chang Cung Memorial Hospital, 215 H/E WSIs, 1.303.012 patches	(a) IR-v2 performed better than the others: AUC 0.97, F-score: 0.97 (b) IR-v2 Type 5: AUC 0.99, F-score: 0.99
	Terradillos [51]	J Pathol Inform	Diagnosis	Two-class classifier based on the Xception model architecture	Binary (benign/malignant)	Basurto University Hospital 14.712 images	SE: 0.8228 ± 0.1575 SP: 0.9114 ± 0.0814
	Paladini [59]	J Imaging	Tumor tissue classification	2 × Ensemble approach ResNet-101, ResNeXt-50, Inception-v3 and DensNet-161. (1) Mean-Ensemble-CNN approach, the predicted class of each image is assigned using the average of the predicted probabilities of four trained models. (2) In the NN-Ensemble-CNN approach, the deep features corresponding to the last FC layer are extracted from the four trained models	1st database: 8-class (tumor epithelium, simple stroma, complex stroma, immune cells, debris, normal glands, adipose tissue, background) 2nd database: 7-class (tumor, complex stroma, stroma, smooth muscle, benign, inflammatory, debris)	Kather-CRC-2016 Database (5000 CRC images) and CRC-TP Database (280,000 CRC images)	Kather-CRC-2016 Database: Mean-Ensemble-CNN mean ACC: 96.16% NN-Ensemble-CNN mean ACC: 96.14% CRC-TP Database: Mean-Ensemble-CNN ACC: 86.97% Mean-Ensemble-CNN F1-Score: 86.99% NN-Ensemble-CNN ACC: 87.26% NN-Ensemble-CNN F1-Score: 87.27%
	Nguyen [110]	Mod Pathol	Consensus molecular subtypes (CMSs) prediction	A system for tissue detection in WSIs based on an ensemble learning method with two raters, VGG and CapsuleNet	Mucin-to-tumor area ratio quantification and binary classification: high/low mucin tumor	TCGA (871 slides) Bern (775 slides) The Cancer Imaging Archive (TCIA) (373 images)	ICC between pathologists and model for mucin-to-tumor area ratio score: 0.92
	Toğaçar [48]	Comput Biol Med	Diagnosis	DarkNet-19 model based on the YOLO object detection model	Binary (benign/colon ADC)	10.000 images	Colon ADC: ACC: 99.96% Colon benign: ACC: 99.96% Overall ACC: 99.69%
	Zhao [95]	Cancer Immunol Immunother	Lymph Node Metastasis (LNM) prediction	Same CNN as Zhao et al., 2020 (VGG-19 pre-trained on the ImageNet using transfer learning with SGDM)	7-class: tumor epithelium, stroma, mucus, debris, normal mucosa, smooth muscle, lymphocytes, adipose	Training 279 H/E WSIs and Validation 194 H/E WSIs	High CLR density OS in the discovery cohort HR: 0.58 High CLR density OS in the validation cohort HR: 0.45
	Kiehl [89]	EJC	Lymph Node Metastasis (LNM) prediction	ResNet18 pre-trained on H&E-stained slides of the CAMELYON16 challenge	Binary (LNM positive/LNM negative)	DACHS cohort (2,431 patients) TCGA (582 patients)	AUROC on the internal test set: 71% AUROC on the TCGA set: 61.2%
	Xu [76]	Caner Cell Int	Quantification of tumor–stroma ratio (TSR) for prognosis	VGG-19 with or w/o transfer learning	9-class: adipose, background, debris, lymphocytes, mucus, muscle, normal mucosa, stroma, tumor epithelium	283.000 H/E tiles, 154.400 IHC tiles from 243 slides from 121 patients, 22.500 IHC tiles from 114 slides from 57 patients	Test dataset: ACC 0.973, 95% CI 0.971–0.975
	Yu [47]	Nat Commun	Diagnosis	No details for deep learning	Binary (cancer/not cancer)	13.111 WSIs, 62,919 patches	Patch-level diagnosis AUC: 0.980 ± 0.014 Patient-level diagnosis AUC: 0.974 ± 0.013
	Jiao [60]	Comput Methods Programs Biomed	Tumor tissue classification	Deep embedding-based logistic regression (DELR), using active learning for sample selection strategy	8-class: adipose, debris, lymphocytes, mucus, smooth muscle, normal mucosa, stroma, tumor epithelium	180.082 patches	AUC: >0.95
	Brockmoeller [90]	J Pathol	Lymph Nodes Metastasis (LNM) prediction	ShuffleNet with transfer learning for end-to-end prediction	(A) Prediction: Any Lymph Node Metastasis (B) >1 lymph node positive	Køge/Roskilde and Slagelse Hospitals/pT2 cohort (311 H/E sections) Retrospective Danish Study/pT1 cohort (203 H/E sections)	pT1 CRC >1 LNM AUROC: 0.733 Any LNM AUROC: 0.567 pT2 CRC >1 LNM AUROC: 0.733 Any LNM AUROC: 0.711
	Mittal [40]	Cancers	Diagnosis	Custom architecture with 12 CN and 3 FC	Binary (cancer/normal)	15 TMAs	ACC:98%, SP: 98.6%, SE: 98.2%
	Kim [114]	Sci Rep	Tumor tissue classification	Combination of InceptionResNet.v2 with PCA and Wavelet transform	5-class: ADC, high-grade adenoma with dysplasia, low-grade adenoma with dysplasia, carcinoid, hyperplastic polyp	Yeouido St. Mary’s Hospital 390 WSIs	Dice: 0.804 ± 0.125 ACC: 0.957 ± 0.025 Jac: 0.690 ± 0.174
	Tsuneki [62]	Diagnostics	Tumor tissue classification	The authors use the EfficientNetB1 model starting with pre-trained weights on ImageNet	4-class: poorly differentiated ADC, well-to-moderately ADC, adenoma, non-neoplastic)	1.799 H/E WSIs	AUC 0.95
	Bustos [106]	Biomolecules	Tumor tissue classification/MSI prediction	Resnet-34 pre-trained on ImageNet	(A) 9-class: adipose, background, debris, lymphocytes, mucus, smooth muscle, normal colon epithelium, cancer-associated stroma, colorectal ADC epithelium (B) Binary (MSI-H/MSS)	72 TMAs	(A) Validation test: AUC 0.98 (B) MSI AUC 0.87 ± 0.03
	Bilal [78]	Lancet Digit Health	Prediction of molecular pathways and mutations	2 × pre-trained models (1) ResNet-18, (2) adaptive ResNet-34	Binary: (1) High/low mutation density (2) MSI/MSS (3) Chromosomal instability (CIN)/Genomic stability (4) CIMP-high/CIMP-low (5) BRAFmut/BRAFWT (6) TP53mut/TP53WT (7) KRASmut/KRASWT	TCGA (502 slides) Pathology Artificial Intelligence Platform (PAIP) challenge—47 slides (12 microsatellite instable and 35 microsatellite stable)	Mean AUROC Hypermutation: (0.81 [SD 0.03] vs. 0.71), MSI (0.86 [0.04] vs. 0.74), CIN (0.83 [0.02] vs. 0.73), BRAF mutation (0.79 [0.01] vs. 0.66), TP53mut (0.73 [0.02] vs. 0.64), KRAS mutation (0.60 [SD 0.04] vs. 0.60), CIMP-high status 0.79 (SD 0.05)
	Nguyen [42]	Sci Rep	Diagnosis	Same approach with Nquyen et al., 2021, presented in Mod Pathol	3-class: Tumor/Normal/Other tissue	54 TMA slides	SVEVC: Tumor: Recall:0.938, Precision:0.976, F1-score: 0.957, ACC: 0.939 Normal: Recall: 0.864, Precision: 0.873, F1-score: 0.915, ACC: 0.982 Other tissue: Recall: 0.964, Precision: 0.772, F1-score: 0.858, ACC: 0.947 Overall (average): Recall: 0.922, Precision: 0.907, F1-score: 0.910, ACC: 0.956
	Shen [46]	IEEE/ACM Trans Comput Biol Bioinform	Diagnosis	A DenseNet based architecture of CNN, in an overall framework which employs a Monte Carlo adaptively sampling to localize patches	3-class: loose non-tumor tissue/dense non-tumor tissue/gastrointestinal cancer tissues	(i) TCGA-STAD 432 samples (ii) TCGA-COAD 460 samples (iii) TCGA-READ 171 samples	DP-FTD: AUC 0.779, FROC 0.817 DCRF-FTD: AUC 0.786, FROC 0.821
2022	Schrammen [79]	J Pathol	Diagnosis/Prediction of IHC biomarkers	Novel method called Slide-Level Assessment Model (SLAM), uses an end-to-end neural network based on ShuffleNet	3-class: Positive tumor slides, Negative tumor slides, Non-tumor slides (A) Binary: BRAF status (mutated or non-mutated) (B) Binary (MSI/MMR) (C) Binary: High grade (grade 3–4)/Low grade (grade 1–2)	(A) Darmkrebs: Chancen der Verhütung durch Screening (DACHS) 2.448 H/E slides B) Yorkshire Cancer Research Bowel Cancer Improvement Program (YCR-BCIP) 889 H/E slides	DACHS cohort Tumor detection AUROC: 0.980 Tumor grading AUROC: 0.751 MSI/MMRD or MSS/MMRP AUROC: 0.909 BRAF status detection AUROC: 0.821 YCR-BCIP cohort MSI/MMRD status detection AUROC: 0.900
	Hosseinzadeh Kassani [43]	Int J Med Inform	Diagnosis	A comparative study between popular architectures (ResNet, VGG, MobileNet, Inceptionv3, InceptionResnetv2, ResNeXt, SE-ResNet, SE-ResNeXt)	Binary (Cancerous/Healthy regions)	DigestPath, 250 H/E WSIs, 1.746 patches	Dice: 82.74% ± 1.77 ACC: 87.07% ± 1.56 F1 score: 82.79% ± 1.79
	Deshpande [116]	Med Image Anal	Diagnosis	Novel GAN architecture, called SAFRON, including loss function which enables generation of images of arbitrarily large sizes after training on relatively small image patches	Binary (benign/malignant)	(A) CRAG (Graham et al., 2019, Awan et al., 2017) 213 colorectal tissue images (B) DigestPath 46 images	ResNet model median classification ACC: 97% with generated images added to the Baseline set, and 93% without

ADC: Adenocarcinoma, ACC: Accuracy, AUC: Area under the ROC Curve, CNN: Convolutional Neural Network, IHC: Immunohistochemistry, SE: Sensitivity, SP: Specificity, TCGA: The Cancer Genome Atlas, SVM: Support Vector Machine, CL: Convolutional layers, FC: Fully-Connected (output) layer, CRC: Colorectal Cancer, TMA: Tissue microarray, WSIs: Whole-slide images, H/E: Hematoxylin and Eosin, MSI: Microsatellite Instability, MMR: Mismatch Repair, MSS: Microsatellite Stable, KRAS: Kirsten rat sarcoma virus, CIN: Chromosomal instability, TP53: Tumor Protein 53, ICOS: Inducible T-cell COStimulator, APC: Adenomatous Polyposis, PIK3CA: Phosphatidylinositol-4,5-Bisphosphate 3-Kinase Catalytic Subunit Alpha.

## 4. Discussion

A pathology diagnosis focuses on the macroscopic and microscopic examination of human tissues, with the light microscope being the valuable tool for almost two centuries [11]. A meticulous microscopic examination of tissue biopsies is the cornerstone of diagnosis and is a time-consuming procedure. An accurate diagnosis is only the first step for patient treatment. It needs to be complimented with information about grade, stage, and other prognostic and predictive factors [4]. Pathologists’ interpretations of tissue lesions become data, guiding decisions for patients’ management. A meaningful interpretation is the ultimate challenge. In certain fields, inter- and intra-observer variability are not uncommon [12,13]. In such cases, the interpretation of the visual image can be assisted by objective outputs. Many data have been published over the last 5 years exploring the possibility of moving on to computer-aided diagnosis and the measurement of prognostic and predictive markers for optimal personalized medicine [117,118]. Furthermore, the implementation of AI is now on the horizon. In the last 5 years, extensive research has been conducted to implement AI-based models for the diagnosis of multiple cancer types and, in particular, CRC [14,15,119]. The important aspects in a CRC diagnosis, such as histological type, grade, stromal reaction, immunohistochemical and molecular features have been addressed using breakthrough technologies.

The traditional pathology methods are accompanied by great advantages [120]. The analytical procedures in pathology laboratories are cost-effective and, during recent years, have become automated, eliminating the time and errors of procedures, while maintaining high levels of sensitivity and specificity of techniques, such as IHC [119]. Despite the widespread availability, challenges and limitations of traditional pathology methods remain, such as the differences between laboratories’ protocols and techniques, as well as the subjective interpretation between pathologists, resulting in inconsistency in diagnoses [12,13]. Novel imaging systems and WSI scanners promise to upgrade traditional pathology, preserving the code and ethics of practice [119]. The potential of DL algorithms is expanding all over the fields in histopathology. In clinical practice, such algorithms could provide valuable information about the tumor microenvironment quantitative analysis of histological features [76]. Better patient stratification for targeted therapies could be approached by DL-based models predicting mutations, such as MSI status [77,78,107]. More than ever, AI could be of great importance for a pathologist in daily clinical practice. AI is consistently supported by extensive research, which is followed by good performance metrics and potential. Several studies have shown that many DL-based models’ predictions did not differ in terms of statistical significance when compared to pathologists’ predictions [45,104]. Thus, DL algorithms could provide valuable results for diagnoses in clinical practice, especially when inconsistencies occur. The available scanned histological images can be reviewed and examined by the collaboration of pathologists simultaneously, from different locations [121,122]. For an efficient fully digital workflow, however, the development of technology infrastructure, including computers, scanners, workstations and medical displays is necessary.

Summarizing the presented DL studies from the medical point of view, 17 studies focus on diagnosis, classifying the images as cancer/not cancer, benign/colon ADC or benign/malignant, 17 studies classify tumor tissues, 19 studies investigate the microenvironment of tumors, 14 studies extract histological features related to prognosis, metastasis and survival, and finally, 10 studies detect the microsatellite instability status. The remaining 5 studies that were not described mainly concerned the technical aspects of DL in histological images of CRC. Summarizing the presented DL works from the technical point of view, 80 studies are applications of CNNs, either for image segmentation or classification, and 2 studies employ GANs for the simulation of histological images. The unbalanced distribution between CNN-based and GAN-based studies is an expected result due to the objectives of these two deep learning approaches. CNNs directly classify the images into different categories (e.g., cancer/not cancer). In contrast, GANs just improve the dataset to avoid overtraining and overfitting during the training procedure, without dealing directly with the main medical question. From the CNN-based studies, 10 studies proposed a custom CNN architecture, which was developed from scratch, 42 studies employed already developed architectures, often using transfer learning, and finally, 26 studies implemented novel architectures, such as (a) the modification of those already developed (5 studies), (b) a combination between CNNs or CNNs with other AI techniques (15 studies) and (c) ensemble methods (6 Studies). Finally, two (2) of the studies did not provide any detail about the DL approach.

The application of DL methods in the diagnosis of CRC over the last 5-years seems to be evolving rapidly, faster than other fields of histopathology. However, it seems that there is an expected gradual evolution, starting from the simple techniques of CNNs, then employing transfer learning to the networks, and finally attempting to develop new architectures, focusing on the requirements of the medical question. Additionally, in the last two years, alternative deep learning techniques such as GANs have started to be used. The contribution of such methods will be significant, since DL requires a sufficient size of the training set to perform well and provide generalization. Large data sets may not always be available from the annotations of pathologists and, therefore, need to be enriched with a simulated training set.

It is expected that CNN’s application directly in histopathological images will present a better performance compared to traditional techniques. CNNs are advantageous over traditional image processing techniques due to the training procedure, while they are also more robust than the traditional AI techniques because they automatically extract features from the image. In this systematic review, different studies use a variety of performance metrics, while the natures of each classification problem are also different to each other. Therefore, it is not meaningful to calculate the average performance value for all the studies. For this reason, only the accuracy (Acc) and area under the curve (AUC), which were used more than the other metrics, have been used to evaluate each different classification problem. The mean value and Standard Error of Mean have been computed for binary classification problems (Acc = 94.11% ± 1.3%, AUC = 0.852 ± 0.066), 3-class classification problems (Acc = 95.5% ± 1.7%, AUC = 0.931 ± 0.051), and finally 8-class classification problems (Acc = 94.4% ± 2.0%, AUC = 0.972 ± 0.022), which provides sufficient samples of these metrics. The above performance values confirm that DL in colorectal histopathological images can achieve a reliable prediction.

## 5. Conclusions

When dealing with human disease, particularly cancer, we need in our armamentarium all available resources, and AI is promising to deliver valuable guidance. Specifically for CRC, it appears that the recent exponentially growing relevant research will soon transform the field of tissue-based diagnoses. Preliminary results demonstrate that AI-based models are further applied in clinical cancer research, including CRC, and breast and lung cancer. However, to overcome several limitations, larger numbers of datasets, quality image annotations, as well as external validation cohorts are required to establish the diagnostic accuracy of DL models in clinical practice. Given the available collected data, a part of the current systematic review could be extended to meta-analysis, especially utilizing the data from retrospective studies and survival analysis. The latter could provide us with a comprehensive status for the contribution of DL methods to the diagnosis of CRC.

## Figures and Tables

**Figure 1 diagnostics-12-00837-f001:**
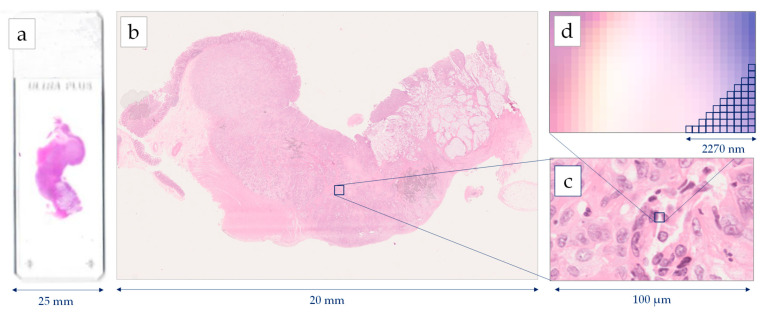
Image generation using a Hamamatsu NanoZoomer whole slide scanner: (**a**) histological slide 75 mm × 25 mm, (**b**) Whole Slide Image (WSI), (**c**) cell level in 40× magnification, (**d**) pixel level in 40× magnification digitizing images 227 nm per pixel.

**Figure 2 diagnostics-12-00837-f002:**
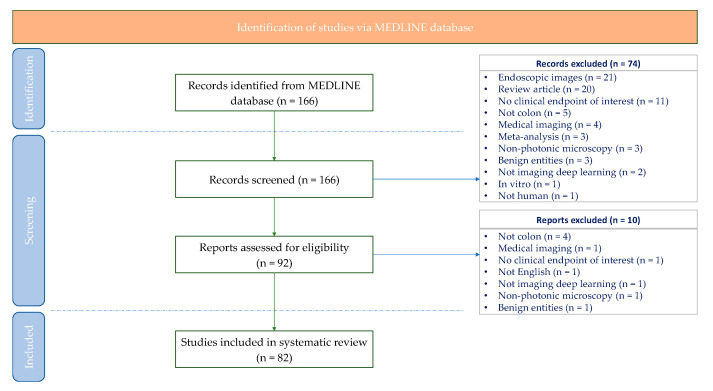
Systematic review flow-chart illustrating systematic search and screening strategy, including number of studies meeting eligibility criteria and number of excluded studies. Last search carried out on 14 January 2022.

**Figure 3 diagnostics-12-00837-f003:**
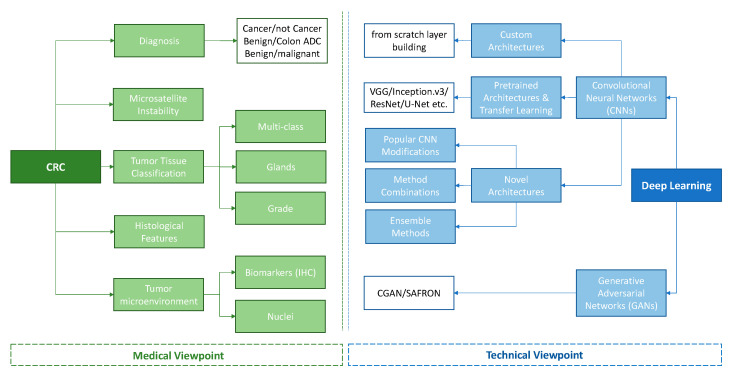
Tree diagram for the categorization of the studies.

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
