# Peer review of "Deep Learning on Histopathological Images for Colorectal Cancer Diagnosis: A Systematic Review"

_diagnostics, 2022, doi:10.3390/diagnostics12040837_

Round 1
Reviewer 1 Report
This is a systematic Review about Deep Learning on Histopathological Images for Colorectal Cancer Diagnosis. The article brings some valuable information regarding the AI role in diagnosis of colorectal cancer. However, please see bellow my suggestions regarding the shape and content of the manuscript.
L39-45 should be supported by refference. I suggest referring/adding Pallag A., Rosca E., Tit D.M., Mutiu G., Bungau S.G., Pop O.L. Monitoring the effects of treatment in colon cancer cells using immunohistochemical and histoenzymatic techniques. Rom. J. Morphol. Embriol., 56(3), 2015, 1103-1109.
L85-90. Please complete the aim of the study with the novelty/special aspects your review brings to the field.
L102. Needing referencing WHO.
L107-122. Too long paragraph not referenced at all. Please revise the entire manuscript and reference EACH statement - this is why you calling it a REVIEW. All statements must be well supported by references.
Figure 2. Please check both Page et al. papers, where this type of graphic is very well described [Page, M.J.; McKenzie, J.E.; Bossuyt, P.M.; Boutron, I.; Hoffmann, T.C.; Mulrow, C.D.; Shamseer, L.; Tetzlaff, J.M.; Akl, E.A.; Brennan, S.E.; et al. The PRISMA 2020 statement: An updated guideline for reporting systematic reviews. Journal of Clinical Epidemiology 2021, 134, 178-189, doi:10.1016/j.jclinepi.2021.03.001. Page, M.J.; McKenzie, J.E.; Bossuyt, P.M.; Boutron, I.; Hoffmann, T.C.; Mulrow, C.D.; Shamseer, L.; Tetzlaff, J.M.; Moher, D. Updating guidance for reporting systematic reviews: development of the PRISMA 2020 statement. Journal of Clinical Epidemiology 2021, 134, 103-112, doi:10.1016/j.jclinepi.2021.02.003} and develop better the flow chart.
Please improve the Discussion chapter by highlighting the benefits of AI diagnosis compared to classic methods of diagnosis (ceva cu cancer de colon). Please describe the modalities of implementation of such systems in hospitals - maybe a summarising scheme would be more relevant.
References should be written in the MDPI style, providing and completing all data requested by the Instructions for authors. Please see https://www.mdpi.com/journal/diagnostics/instructions and proceed.
Author Response
Response for the manuscript entitled “Deep Learning on Histopathological Images for Colorectal Cancer Diagnosis: A Systematic Review”
Reviewer #1
We thank the reviewer for his valuable contribution. We have addressed all the reviewer’s comments as described below:
Comment 1: L39-45 should be supported by refference. I suggest referring/adding Pallag A., Rosca E., Tit D.M., Mutiu G., Bungau S.G., Pop O.L. Monitoring the effects of treatment in colon cancer cells using immunohistochemical and histoenzymatic techniques. Rom. J. Morphol. Embriol., 56(3), 2015, 1103-1109.
Response 1: We have included the proposed reference in the introduction section (line 46, ref [8])
Comment 2: L85-90. Please complete the aim of the study with the novelty/special aspects your review brings to the field.
Response 2: The last paragraph of the introduction section has been modified and some text has been added to make clearer the novelty of the review.
Comment 3: L102. Needing referencing WHO.
Response 3: We have included the reference of WHO (ref [16]).
Comment 4: L107-122. Too long paragraph not referenced at all. Please revise the entire manuscript and reference EACH statement - this is why you calling it a REVIEW. All statements must be well supported by references.
Response 4: Five (5) references have been included to support all the statements of the Paragraph. Furthermore, twenty-three (23) more references have been added in the manuscript in total.
Comment 5: Figure 2. Please check both Page et al. papers, where this type of graphic is very well described [Page, M.J.; McKenzie, J.E.; Bossuyt, P.M.; Boutron, I.; Hoffmann, T.C.; Mulrow, C.D.; Shamseer, L.; Tetzlaff, J.M.; Akl, E.A.; Brennan, S.E.; et al. The PRISMA 2020 statement: An updated guideline for reporting systematic reviews. Journal of Clinical Epidemiology 2021, 134, 178-189, doi:10.1016/j.jclinepi.2021.03.001. Page, M.J.; McKenzie, J.E.; Bossuyt, P.M.; Boutron, I.; Hoffmann, T.C.; Mulrow, C.D.; Shamseer, L.; Tetzlaff, J.M.; Moher, D. Updating guidance for reporting systematic reviews: development of the PRISMA 2020 statement. Journal of Clinical Epidemiology 2021, 134, 103-112, doi:10.1016/j.jclinepi.2021.02.003} and develop better the flow chart.
Response 5: Figure 2 has been improved according to the proposed works and the updated PRISMA guidance.
Comment 6: Please improve the Discussion chapter by highlighting the benefits of AI diagnosis compared to classic methods of diagnosis (ceva cu cancer de colon). Please describe the modalities of implementation of such systems in hospitals - maybe a summarising scheme would be more relevant.
Response 6: A paragraph (Page 28, Lines 571-593) has been included to highlight the benefits of AI diagnosis compared to classic methods of diagnosis.
Comment 7: References should be written in the MDPI style, providing and completing all data requested by the Instructions for authors. Please see https://www.mdpi.com/journal/diagnostics/instructions and proceed.
Response 7: The format of all the references has been corrected using the Mendeley reference manager.
Reviewer 2 Report
I appreciate reading your interesting article discussing the AI model developing newly day by day. You introduced many fancy studies using the DL model in CRC. Can you explain why the CNN model was applied much more than the GAN and Ensemble model in CRC? Is the CNN model is superior to the rest, especially in CRC? How about other types of cancer? If possible, please summarize the comments I mentioned in your conclusion.
Author Response
Response for the manuscript entitled “Deep Learning on Histopathological Images for Colorectal Cancer Diagnosis: A Systematic Review”
Reviewer #2
Comment: I appreciate reading your interesting article discussing the AI model developing newly day by day. You introduced many fancy studies using the DL model in CRC. Can you explain why the CNN model was applied much more than the GAN and Ensemble model in CRC? Is the CNN model is superior to the rest, especially in CRC? How about other types of cancer? If possible, please summarize the comments I mentioned in your conclusion.
Response: We thank the reviewer for his appreciation. It is indeed extremely unbalanced the number of CNN based studies compared to the GANs ones. However, this is an expected result, due to the objectives of these two deep learning approaches. CNNs directly classify the images into different categories (e.g. cancer/not cancer). In contrast, GANs just improves the dataset to avoid overtrain and overfitting during the training procedure, without dealing directly with the main medical question This is the reason we only two studies have been presented using GANs. We add some texts (Page 28, Lines 602-607) to clarify this issue.
Round 2
Reviewer 1 Report
All my requirements have been addressed.